# Programmable Assembly of Multistranded Helices in Water

Dimitri Delcourt [1], Reguram Arumugaperumal[1], Prachi Verma[1], Perttu Permi [1,2], Rosa M. Gomila [3], Antonio Frontera [3] & Fabien B. L. Cougnon [1]✉

Sequence-specific conformational changes underpin essential biological processes, from information storage to energy transduction, but are difficult to replicate in synthetic systems. Here, we present a simple approach to encode in the primary sequence of molecular strands all the information required to govern both the formation and dynamic behavior of multistranded helices. We demonstrate that the sequence of oligo(*m*-phenylene ethynylene) strands composed of hydrophobic phenylene and charged pyridinium residues reliably direct the formation of either a single structure (e.g., a double helix) or dynamic assemblies (e.g., double and triple helices in exchange). In the latter case, transitions between different helical states can be controlled by concentration, temperature, or by the presence of anionic molecules. This minimal yet versatile design strategy lays the groundwork for the construction of adaptive supramolecular systems with programmable structure and function.

The primary sequence of proteins and nucleic acids defines a landscape of energetically accessible conformational states, enabling these biomolecules to undergo controlled structural transitions in response to external cues[1]. The primary sequence of DNA, for example, determines its ability to adopt a variety of higher-order structures beyond the canonical right-handed B-form, including left-handed double helices (Z-form), triple helices (H-form), and G-quadruplexes[2,3]. Transitions between these different states play essential roles in key cellular processes like gene expression, replication, and repair, showing that precise sequence control governs not only the dynamic behavior of biomolecules but, ultimately, their function.

Significant efforts have been devoted to developing synthetic analogs that mimic the structural and functional complexity of biomolecules. Synthetic multistranded helices have emerged as promising biomimetic platforms for applications such as molecular recognition[4–10], switching[11–17], and self-replication[18,19]. Their assembly typically relies on non-covalent interactions, including hydrogen bonding[20,21], π–π stacking[22–29], and metal coordination[13,30,31]. Despite the wide range of strategies available to control helix formation, general methods for encoding both structural organization and dynamic responsiveness directly into the primary sequence of synthetic strands

have yet to be established. In a landmark study, the Flood group demonstrated sequence control over single- to double-helix transition in aryl-triazole foldamers[26]. However, extending this approach beyond that specific system remains an open challenge. Developing more broadly applicable strategies that couple sequence information to both structure and dynamics would unlock a new generation of synthetic systems with biomolecule-like functions.

Here we present an approach for encoding, within the primary sequence of synthetic strands, all the information necessary to control both the assembly and dynamic behavior of multistranded helices. Figure 1a shows the general structure of a bolaamphiphilic strand, consisting of a central hydrophobic core flanked by two permanently charged, polar termini. Let $h$ and $p$ denote the lengths of the hydrophobic and polar segments, respectively. Such bolaamphiphilic sequences promote the self-assembly of $n$-stranded helices ($n = 1, 2, 3, ...$) in aqueous environments. As illustrated in Fig. 1b, the resulting helices feature a central hydrophobic layer flanked by two charged layers at the upper and lower rims, effectively minimizing exposure of the hydrophobic surface to water.

The formation of these helices occurs only if the charged segments do not stack directly above one another, as this would

[1]Department of Chemistry, Nanoscience Center, University of Jyväskylä, JYU, Finland. [2]Department of Biological and Environmental Science, Nanoscience Center, University of Jyväskylä, JYU, Finland. [3]Department de Química, Universitat de les Illes Balears, Carretera de Valldemossa km 7.5, Palma de Mallorca, Baleares, Spain. ✉e-mail: fabien.b.l.cougnon@jyu.fi

**a**

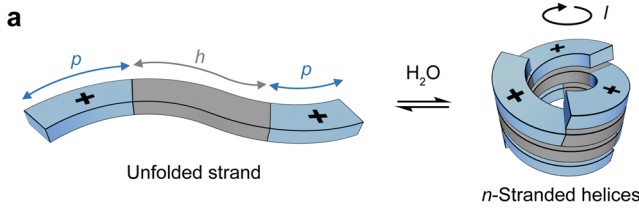

**b**

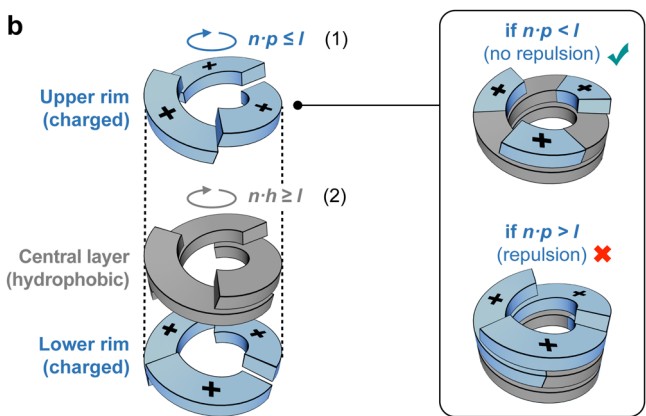

Fig. 1 | **Sequence-encoded assembly of multistranded helices from bolaam-
phiphilic molecular strands in water. a** Schematic representation of a bolaam-
phiphilic strand assembling into $n$-stranded helices in water. The hydrophobic core
length is denoted by $h$, the length of the polar, permanently charged termini by $p$,
and the length of strand require to complete one helical turn by $l$. **b** Cutaway view of
a helix illustrating how geometric constraints, set by the primary sequence of the
strand, determine the type of helix that forms. The inset illustrates how variations in
the length of the permanently charged termini located at the upper rim can result in
electrostatic repulsion (the lower rim of the helix is not shown for clarity).

generate destabilizing electrostatic repulsion. Repulsion between
charged segments located on the upper rim can be avoided if the
combined length of these charged segments ($n·p$) does not exceed
the length of one full helical turn (Fig. 1b, inset). Defining $l$ as the
length of strand required to complete one helical turn, this condition
can be expressed as:

$$n·p ≤ l \qquad (1)$$

By symmetry, this same constraint applies to the charged seg-
ments located at the lower rim of the helix.

Repulsion between charged segments located on opposite rims of
the helix can be similarly avoided if they are separated by a full
hydrophobic layer; that is, if the combined length of hydrophobic
components located at the center of the structure ($n·h$) is greater or
equal to $l$:

$$n·h ≥ l \qquad (2)$$

Combining inequalities (1) and (2) yields a single inequality that
defines the permissible range for the number of strands $n$:

$$l/h ≤ n ≤ l/p \qquad (3)$$

This inequality establishes a direct sequence-to-structure rela-
tionship: the primary sequence of the strand (characterized by para-
meters $h$, $p$, and $l$) sets the geometric window within which $n$-stranded
helices are stable. When $n$ falls within this window, helix formation is
thermodynamically favored, as it reduces hydrophobic exposure

without introducing repulsive interactions. Outside this range, helix
formation is disfavored due to unavoidable electrostatic destabiliza-
tion. Inequality (3) can thus be used to parametrize strands capable of
forming specific $n$-stranded helices.

## Results

In this article, we sought to experimentally test the predictions of the
model described above. Real molecular systems do not necessarily
meet the assumptions of this idealized framework, which presumes a
backbone of constant thickness, sufficient flexibility to accommodate
helices of any multiplicity, and uniformly distributed charges across
the polar termini. Testing the validity of our model therefore required
choosing a molecular scaffold that approximates these idealized fea-
tures as closely as possible.

Oligo($m$-phenylene ethynylene)s were selected as a suitable
scaffold, because their conformational behavior is well
documented[32–37]. Moore demonstrated their strong tendency to fold
into single helices several decades ago[38–40]. The folding process is
driven by solvophobic effects and can be promoted by polar solvents
such as water[41]. More recently, Berryman showed that oligo($m$-phe-
nylene ethynylene)s can also form higher-order helices[42–44]. Building
on these foundations, we reengineered the oligo($m$-phenylene ethy-
nylene) scaffold by introducing hydrophobic and polar segments of
variable lengths, and assessed whether the resulting sequences pre-
dictably yielded specific types of multistranded helices.

### Controlling the assembly of a single type of multistranded helix
We first designed strand **1** (Fig. 2a), composed of $h = 3$ hydrophobic
phenylene residues and $p = 3$ polar pyridinium residues. Since oligo($m$-
phenylene ethynylene) helices require $l = 6$ aromatic residues per
turn[38–44], the lower and upper bounds of inequality (3) are $l/h = 2$ and
$l/p = 2$, respectively. Because $n = 2$ is the only integer that satisfies this
condition, strand **1** is predicted to exclusively assemble into a dou-
ble helix.

This situation can be schematically represented with a diagram
(Fig. 2b) comparing the relative enthalpies of three possible states:
$n = 1$ (single helix), $n = 2$ (double helix), and $n = 3$ (triple helix). The
double helix is more stable (lower enthalpy) than the single and triple
helices (higher enthalpy) because it is the only configuration that
avoids electrostatic repulsion. It should be noted that these enthalpic
levels are arbitrary and intended solely as a qualitative illustration of
the allowed and forbidden states.

The equilibria involved in the folding and assembly of strand **1** are
shown in Fig. 2c. According to our predictions, single helix formation is
disfavored, as it would bring charged pyridiniums in close contact,
resulting in destabilizing electrostatic repulsion. The formation of a
double helix (which was modeled using a BP86-D4-COSMO(water)/
def2-TZVP level of theory) allows the outer pyridiniums to effectively
shield the hydrophobic phenylenes from water. The assembly of
higher-order helices (not shown in Fig. 2c) is also prohibited, since the
pyridinium residues already occupy all available positions at both the
top and the bottom of the double helix, making it impossible for
additional charged residues to fit without overlapping.

Strand **1** was synthesized in ten steps (overall yield: 5.8%) using
iterative palladium-catalyzed Sonogashira cross-coupling reactions, as
detailed in the Supplementary Information. The final product was
isolated with either trifluoroacetate (**1·TFA**) or triflate (**1·OTf**)
counterions.

Unfolded strand **1·TFA** was first characterized by $^1$H and $^{13}$C NMR
spectroscopy in CD$_3$CN (Figures S3-S5), a moderately polar solvent
that efficiently solvates the strand and does not promote folding.
Under these conditions, **1·TFA** displayed sharp signals (Fig. 2d) con-
gruent with a discrete monomeric species with a hydrodynamic radius
$r_H = 12.0$ Å, estimated by $^1$H diffusion-ordered spectroscopy (DOSY,
Figure S6) using the Stokes-Einstein equation.

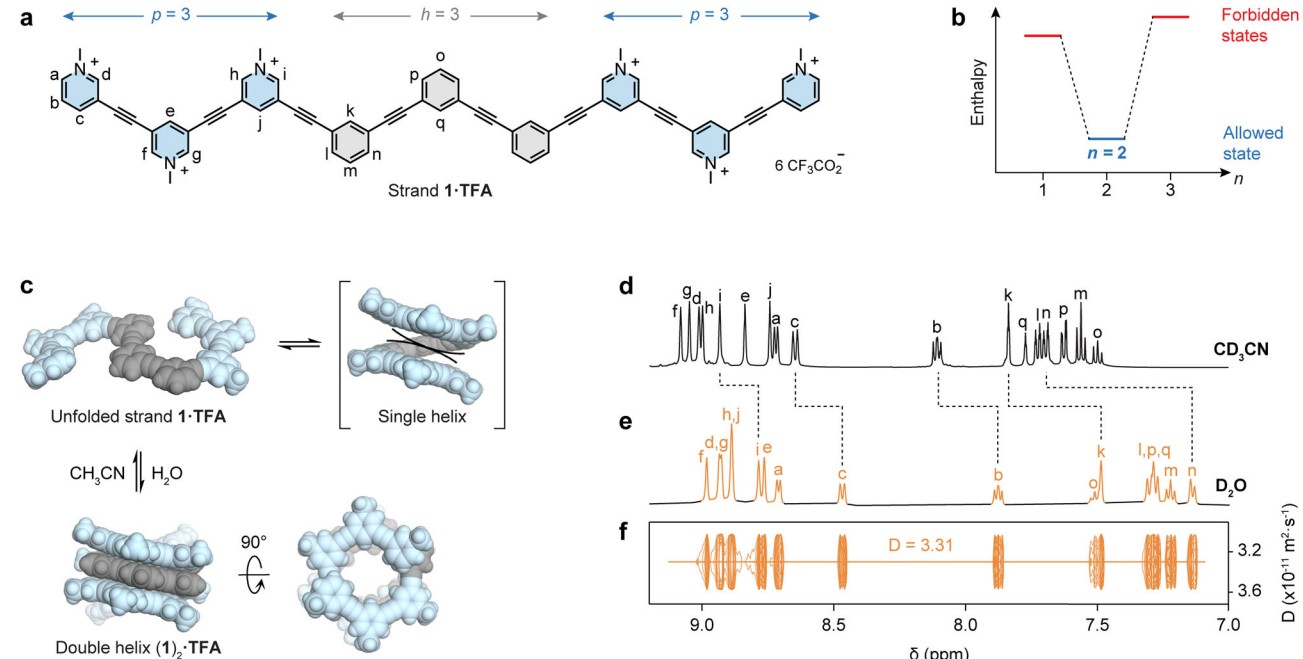

**Fig. 2 | Programmable assembly of a double helix in aqueous solution.**
**a** Chemical structure of strand **1·TFA** (TFA: trifluoroacetate). The variables $h$ and $p$ denote the lengths of the hydrophobic core and the polar termini, respectively. **b** Energy diagram showing that the double helix is the only thermodynamically favored state in water. The variable $n$ denotes the number of strands forming each helix. **c** Equilibria involved in the formation of double helix $(1)_2$. Double helix $(1)_2$ was modeled using a BP86-D4-COSMO(water)/def2-TZVP level of theory. **d** ¹H NMR spectrum of unfolded strand **1·TFA** in CD₃CN (5 mM, 500 MHz, 298 K). **e** ¹H NMR and **f** DOSY spectra of double helix $(1)_2$·**TFA** in D₂O (10 mM, 500 MHz, 298 K). Symbols δ and D denote the chemical shift and diffusion coefficient, respectively.

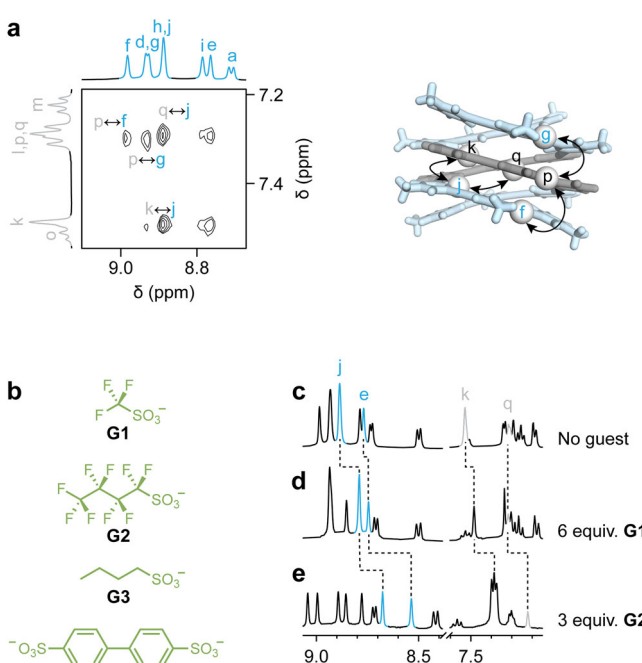

**Fig. 3 | Structural characterization and host-guest chemistry of double helix $(1)_2$. a** Partial 2D ¹H-¹H ROESY NMR spectrum of double helix $(1)_2$·**TFA** in D₂O (10 mM, 500 MHz, 278 K, 100 ms mixing time), showing key through-space correlations between protons of stacked pyridinium (blue) and phenylene (gray) residues. **b** Chemical structure of hydrophobic sulfonates **G1-G4**. **c**–**e** Partial ¹H NMR spectra of double helix $(1)_2$·**TFA** in D₂O (5 mM in D₂O, 500 MHz, 298 K): **c** free host, **d** after addition of 6.0 equiv. of **G1**, **e** after addition of 3.0 equiv. of **G2**. The symbol δ denotes the chemical shift.

Substantial spectral changes, indicative of duplex formation, were observed upon addition of D₂O (Figures S7). In pure D₂O (Fig. 2e), the phenylene protons exhibited pronounced upfield shifts (e.g., Δδ = 0.3 ppm for proton k, Δδ = 0.5 ppm for proton n), consistent with increased shielding due to π-stacking interactions. Moreover, DOSY measurements conducted in D₂O (Fig. 2f) yielded a hydrodynamic radius $r_H = 8.4$ Å, well aligned with the dimensions of the optimized DFT model for the double helix (ca. 3.9 Å in height and 9.5 Å in width). The reduction in hydrodynamic radius from the unfolded strand (12.0 Å) to the assembled duplex (8.4 Å) supported the formation of a more compact structure, closer to the spherical shape assumed in the Stokes-Einstein equation.

Additional evidence for duplex formation came from 2D ¹H-¹H rotating frame Overhauser effect spectroscopy (ROESY). The ROESY spectrum (Fig. 3a) revealed multiple through-space ROE correlations between protons of stacked phenylene and pyridinium residues (e.g., p ↔ g and p ↔ f ouside the helix, and k ↔ j and q ↔ j inside the helix). These cross-peaks would not be observable in a single strand (folded or not), as the corresponding protons would have been separated by distances of at least ~8 Å. A more detailed analysis of 2D NMR spectra is provided in the Supplementary Information (Figures S8-S9).

The duplex exhibited remarkable stability in water. It retained its structural integrity upon dilution to concentrations as low as 0.5 mM (Figure S10) and resisted thermal denaturation at temperatures up to 348 K (Figures S11-S12). Duplex formation was also found to be independent of the counterions tested (trifluoroacetate or triflate). However, subtle differences in the NMR spectra of $(1)_2$·**TFA** and $(1)_2$·**OTf** (Figure S13) revealed that the double helix interacts differently with these two counterions.

Triflate, being more chaotropic than trifluoroacetate[45], is more likely to associate with the helix. This consideration led us to investigate the interaction between $(1)_2$·**TFA** and triflate **G1** (Fig. 3b), as well as other anionic organic molecules (**G2-G4**). Titrating potassium triflate

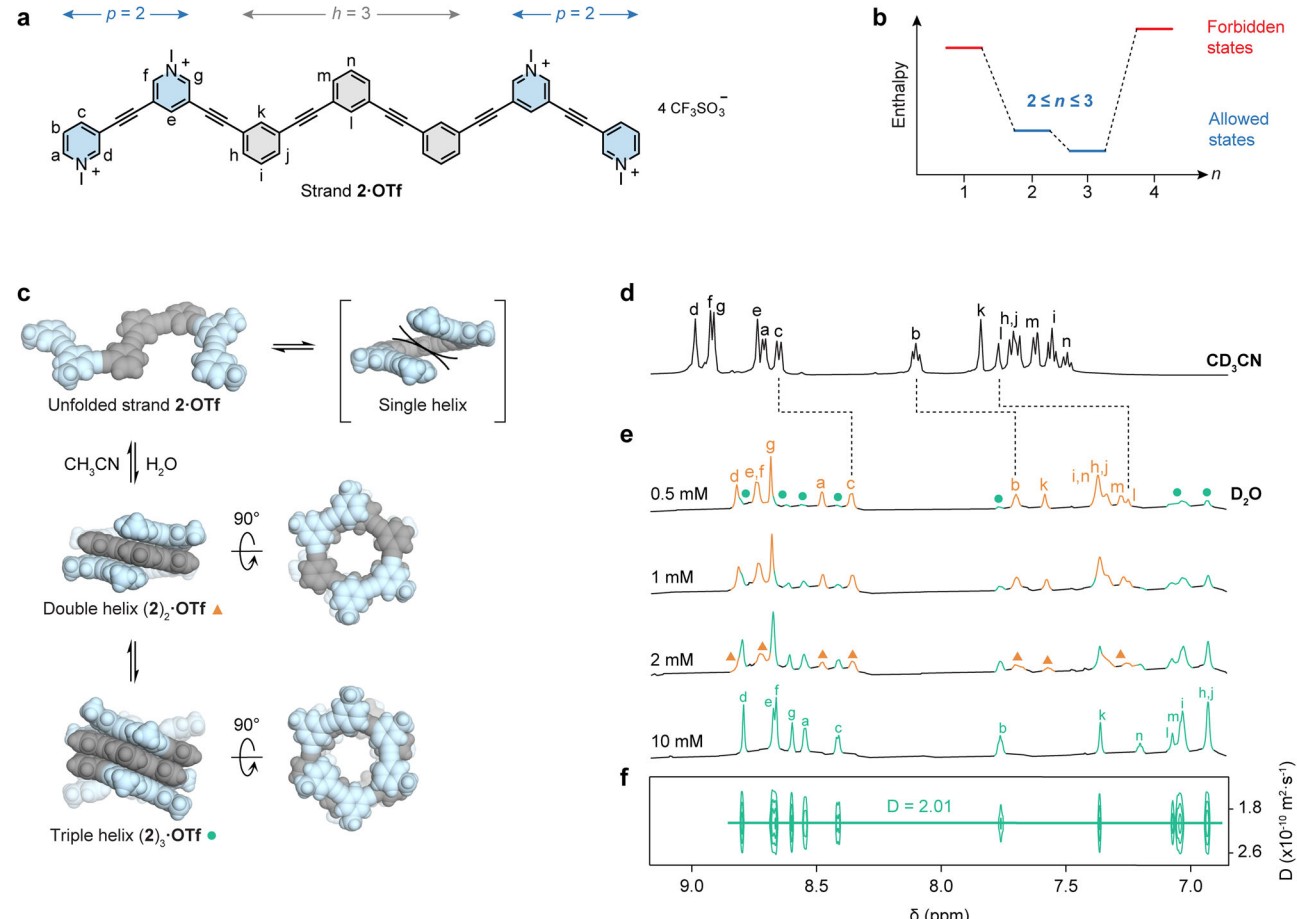

**Fig. 4 | Programmable assembly of double and triple helices in dynamic exchange in aqueous solution. a** Chemical structure of strand **2·OTf** (OTf: triflate). The variables $h$ and $p$ denote the lengths of the hydrophobic core and the polar termini, respectively. **b** Energy diagram indicating that the double and triple helices are the only thermodynamically favored state in water. The variable $n$ denotes the number of strands forming each helix. **c** Equilibria governing the formation of double helix $(2)_2$ and triple helix $(2)_3$. Double helix $(2)_2$ and triple helix $(2)_3$ were both modeled using a BP86-D4-COSMO(water)/def2-TZVP level of theory. **d** $^1$H NMR spectrum of unfolded strand **2·OTf** in CD$_3$CN (10 mM, 500 MHz, 298 K). **e** Concentration-dependent $^1$H NMR spectra showing the formation of double helix $(2)_2$·**OTf** and triple helix $(2)_3$·**OTf** in D$_2$O (800 MHz, 288 K). **f** DOSY spectrum of triple helix $(2)_2$·**OTf** in D$_2$O (10 mM, 800 MHz, 288 K). Symbols δ and D denote the chemical shift and diffusion coefficient, respectively.

**G1** (Fig. 3c, d) into a 5 mM aqueous solution of **1·TFA** induced relatively small shifts in the phenylene proton signals. More pronounced shifts in the pyridinium protons indicated preferential interaction with the polycationic outer rim of the helix. Assuming the formation of a 1:1 complex, analysis of the titration data yielded a relatively modest association constant of $K_a = (75 \pm 2)$ M$^{-1}$ (Figure S14).

We hypothesized that longer, more hydrophobic anions **G2-G4** could bind more strongly by threading through the helix cavity (internal volume: 378 Å$^3$). Indeed, addition of potassium perfluorobutanesulfonate (**G2**) induced significant NMR shifts of both inner pyridinium and phenylene protons, consistent with the formation of an inclusion complex (Fig. 3e and Figure S15). The enhanced binding affinity $K_a = (5.7 \pm 0.4) \times 10^3$ M$^{-1}$ was attributed to the increased hydrophobicity of **G2** and its shape complementarity with the helix cavity. In contrast, the non-perfluorinated analogue sodium butanesulfonate **G3**, which is less hydrophobic, showed no detectable interaction with the double helix (Figure S16). Finally, addition of sodium 4,4′-biphenyldisulfonate (**G4**) resulted in substantial signal broadening, indicative of duplex dissociation, and ultimately led to precipitation (Figure S16).

Overall, these results underscore the predictive accuracy of the sequence-to-structure relationship defined by inequality (3). The thermodynamic stability of the resulting double helix is exceptional, especially given the absence of inter-strand hydrogen bonding or metal coordination. Furthermore, the presence of a hydrophobic cavity within the helix enables the formation of host-guest complexes. One of these guests, perfluorobutanesulfonate **G3**, is a persistent pollutant likely to be banned globally in the near future[46,47], suggesting that oligo($m$-phenylene ethynylene) helices could serve as platforms for capturing such contaminants from water.

### Controlling the assembly of multiple multistranded helices

Encouraged by this initial success, we sought to determine whether our approach could be extended to access more complex, dynamic systems. To this end, we designed strand **2** (Fig. 4a), composed of $h = 3$ hydrophobic phenylene residues flanked by $p = 2$ polar pyridinium residues on each side. Given that $l = 6$ aromatic residues per turn for oligo($m$-phenylene ethynylene) helices, the lower and upper bounds of inequality (3) are now $l/h = 2$ and $l/p = 3$, respectively. These values indicate that strand **2** can assemble into both double and triple helices. The corresponding schematic enthalpic diagram is showed in Fig. 4b, and the equilibria involved are depicted in Fig. 4c.

This system is dynamic in nature, as it permits reversible switching between two distinct, thermodynamically stable helical states. Importantly, this dynamic behavior remains tightly constrained by the boundary conditions imposed by inequality (3). As a result, switching between double and triple helices occurs without any risk of forming

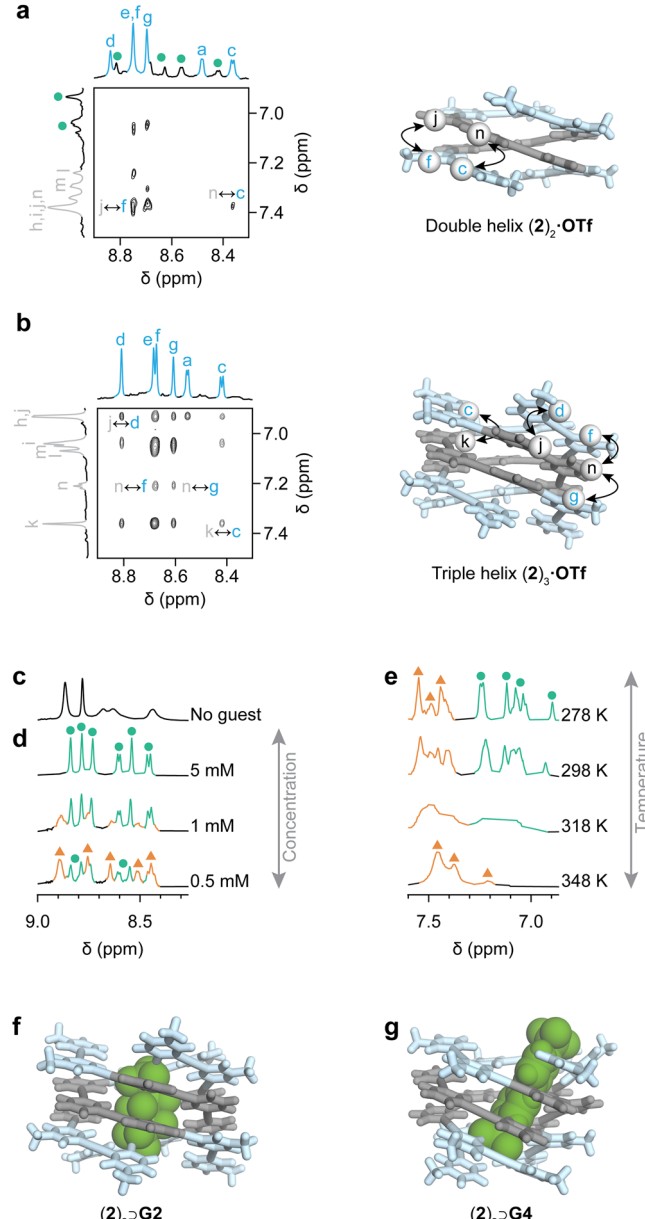

**Fig. 5 | Characterization and dynamic behavior of double helix (2)₂ and triple helix (2)₃. a, b** Partial 2D ¹H-¹H ROESY NMR spectra of **a**, double helix (2)₂·OTf in D₂O (0.5 mM, OTf: triflate) and **b** triple helix (2)₃·OTf (10 mM) in D₂O (800 MHz, 288 K, 200 ms mixing time), highlighting key through-space correlations between protons of stacked pyridinium (blue) and phenylene (gray) residues. **c** ¹H NMR spectrum of triple helix **2·TFA** in D₂O (5 mM, 500 MHz, 298 K). **d** Concentration-dependent ¹H NMR spectra of **2·TFA** (5 to 0.5 mM) in D₂O (500 MHz, 298 K) in the presence of 5.0 equiv. of potassium perfluorobutanesulfonate **G2**. **e** Variable temperature ¹H NMR spectra of **2·TFA** recorded between 278 K and 348 K in D₂O (0.5 mM, 500 MHz) in the presence of 5.0 equiv. of potassium per-fluorobutanesulfonate **G2**. Signals corresponding to the double and triple helices are labelled with orange triangles and green circles, respectively. **f** DFT minimized model of (2)₃⊃**G2** [BP86-D4-COSMO(water)/def2-TZVP]. **g** DFT minimized model of (2)₃⊃**G4** [BP86-D4-COSMO(water)/def2-TZVP]. The symbol δ denotes the chemical shift.

the simpler single helix, a configuration typically favored in related systems[25–27,48,49], but precluded here because $n = 1$ lies outside the predicted stability window.

Strand **2** was synthesized in seven steps with an overall yield of 6.1% (see Supplementary Information), and its assembly was investigated using the protocol described above. In this case, the behavior of

**2·OTf** and **2·TFA** differed markedly. This section focusses on **2·OTf**; the behavior of **2·TFA** will be discussed in the following section.

As expected, the NMR spectrum of **2·OTf** in CD₃CN (Fig. 4d) exhibited sharp signals, consistent with a monomeric species ($r_H = 10.8$ Å, Figure S17), which was fully characterized by ¹H and ¹³C NMR (Figure S18-S20). The signals progressively broadened upon gradual addition of D₂O (Figure S21). This transformation culminated in pure D₂O with the appearance of two new sets of resonances (Fig. 4e and Figure S22-S23), which were assigned to double helix (2)₂ (orange triangles) and triple helix (2)₃ (green circles) based on their chemical environments, hydrodynamic radii and ROE correlations.

Double and triple helices could be individually characterized, because they exchange slowly on the NMR timescale and display well-resolved, distinguishable spectral features. In accordance with Le Chatelier's principle, their relative abundance was found to be concentration dependent. Double helix (2)₂ was predominant at low concentrations (0.5 mM). Its resonances were upfield shifted relative to the unfolded strand and its hydrodynamic radius ($r_H = 8.1$ Å, Figure S24) closely matched that previously measured for double helix (1)₂.

The phenylene proton resonances of triple helix (2)₃ were shifted even further upfield (up to $\Delta\delta = 0.5$ ppm, compared to the double helix signals), as expected from a structure with an increased number of stacked aromatic surfaces. The triple helix was present only in trace amounts at low concentration (0.5 mM) but became the main species observable at 5 mM and above. Its hydrodynamic radius ($r_H = 9.4$ Å, Fig. 4f) was larger than that of the double helix and in excellent agreement with the dimensions of the corresponding DFT model (ca. 5.8 Å height by 9.5 Å width).

Two-dimensional ROESY spectra recorded at concentrations favoring either the double helix (0.5 mM, Fig. 5a) or the triple helix (10 mM, Fig. 5b) further supported our structural assignment. The two helices could be distinguished from the relative spatial arrangement of phenylene and pyridinium residues. For example, phenylene proton n showed through-space correlations with pyridinium proton c in the double helix, but with pyridinium protons f and g in the triple helix. Likewise, phenylene proton j correlated with pyridinium proton f in the double helix, but with pyridinium proton d in the triple helix. Additional diagnostic 2D correlations are presented in Figures S25-S28.

These results further validate the predictive power of our sequence-to-structure model and demonstrate that bolaamphiphilic strands can be programmed to form multiple helical architectures in reversible exchange.

### Controlling the conformational switch between double and triple helices: role of concentration, temperature, and anions

Having established that strand **2·OTf** assembles into double and triple helices in dynamic equilibrium, we examined how external parameters influenced this conformational switch. Specifically, we investigated the effects of concentration, temperature, and anionic guests on the relative populations of the two helical states.

In general, triple helix (2)₃ was favored at higher concentrations and lower temperatures, while double helix (2)₂ predominated under dilute conditions and elevated temperatures. Varying both parameters thus provided a reliable means to control the composition of the system. For instance, at 10 mM, the triple helix remained stable across a broad temperature range (288–328 K, Figure S29). In contrast, at 3 mM, increasing the temperature from 288 K to 328 K induced a clear transition from the triple to the double helix (Figure S30).

We next investigated whether the double-to-triple helix transition could be influenced by guest binding. As a starting point for this study, we compared the behaviors of strands **2·TFA** and **2·OTf**, which differ only in their counterions. The ¹H NMR spectrum of **2·TFA** at 5 mM in D₂O (Fig. 5c) displayed upfield-shifted resonances characteristic of the triple helix, but these signals were significantly broader than those

previously observed for **2·OTf** (Fig. 4e). This difference suggested that the more chaotropic triflate occupies, at least partially, the cavity of the triple helix, thereby restricting its conformational motion. This interpretation was supported by the observation that adding potassium triflate **G1** to a 5 mM aqueous solution of **2·TFA** led to progressive sharpening and shifting of the triple helix resonances (Figure S31). The shifts were more pronounced for inner phenylene protons, as expected if triflate binds within the cavity. Fitting the titration data using a 1:1 binding model yielded an association constant of $K_a = (9.8 \pm 0.7) \times 10^2 \, \mathrm{M}^{-1}$, approximately one order of magnitude higher than that previously measured for $(\mathbf{1})_2$. This increased affinity is attributed to the longer and more hydrophobic internal cavity of the triple helix $(\mathbf{2})_3$ (internal volume: $469 \, \text{Å}^3$), which provides a more favorable environment for guest encapsulation.

Similar signal sharpening was observed upon addition of potassium perfluorobutanesulfonate (**G2**, Figure S32) and sodium 4,4'-biphenyldisulfonate (**G4**, Figures S33-S34), indicating the formation of inclusion complexes $(\mathbf{2})_3 \supset \mathbf{G2}$ and $(\mathbf{2})_3 \supset \mathbf{G4}$, respectively. Although the broad resonances at intermediate titration points precluded accurate determination of association constants for both guests, the importance of hydrophobicity in guest binding was highlighted once again by the lack of measurable interaction between sodium butanesulfonate **G3** and the triple helix (Figure S35).

Concentration- and temperature-dependent NMR spectra recorded after addition of **G1, G2** and **G4** (Figures S36-S41) demonstrated a clean transition between the double and triple helical states in the presence of these guests. A representative example of these experiments, obtained after addition of **G2**, is shown in Fig. 5d, e. Comparisons between the effects of different anions on the switching behavior should be made with caution, as they depend on both the association constants with the double and triple helix, and the number of guest equivalents added. Nevertheless, the data suggest that more hydrophobic guests, expected to bind more strongly to the triple helix, stabilize it and make its conversion into the double helix less favorable both upon dilution and heating. The most stable complexes, $(\mathbf{2})_3 \supset \mathbf{G2}$ and $(\mathbf{2})_3 \supset \mathbf{G4}$, were modeled by DFT at the BP86-D4-COSMO(water)/def2-TZVP level of theory (Fig. 5f, g), confirming a good geometric fit of these guests within the cavity of the triple helix. In the case of **G4**, π-π stacking and electrostatic attraction between the aryl sulfonate moieties and the pyridinium units induces dynamic distortions of the triple helix, consistent with the slightly broader NMR signals obtained for the corresponding complex.

Taken together, these results demonstrate that three external parameters (concentration, temperature, and the presence of suitable anionic guests) can be used to reversibly control the conformational equilibrium between double and triple helices. Importantly, the switching process remained tightly controlled and cleanly operable under all tested conditions, with no detectable formation of the undesired single helices or strand unfolding.

## Discussion

Two bolaamphiphilic oligo(*m*-phenylene ethynylene) strands were synthesized to demonstrate that the relative lengths of hydrophobic and charged segments dictate the formation of distinct helical architectures. As predicted, strand **1** assembled exclusively into a stable double helix, whereas strand **2** formed both double and triple helices in dynamic equilibrium. The latter system enabled clean, reversible switching between the permitted helical states in response to concentration, temperature, or binding of specific amphiphilic guests.

This study establishes a robust strategy for controlling both the assembly and dynamic behavior of multistranded helices through sequence design in oligo(*m*-phenylene ethynylene) strands. The key strength of this approach lies in its minimalism: precise structural control was achieved using only two building blocks, hydrophobic phenylene and charged pyridinium residues. This simplicity renders

the approach modular and potentially extensible to alternative sequence patterns, including non-bolaamphiphilic ones. The strategy may also be applied to strands containing anionic residues (instead of cationic ones) and to other polyaromatic backbones (such as *ortho*-phenylene ethynylene and related *ortho*- or *meta*-phenylene frameworks) provided they possess sufficient flexibility to form multi-stranded helices. Moreover, the oligo(*m*-phenylene ethynylene) backbone is amenable to chemical modification[38,41], allowing for the integration of catalytic, recognition, or sensing functions within structurally programmed helices.

More broadly, the ability to encode multiple, switchable helical states within a single strand introduces a new level of control in the design of adaptive molecular systems. By establishing a clear and predictable sequence-to-structure relationship, this work may enable the construction of complex, programmable supramolecular architectures with responsive functionalities.

## Methods

All reagents and solvents were purchased from commercial sources and used without further purification – except triethylamine, which was distilled on potassium hydroxide under inert atmosphere before being used.

### NMR analyses

Spectra were measured on a JEOL 400 MHz NMR ECX-400 spectrometer equipped with a 5-mm broadband 40TH5AT/FG2 autotunable universal probe, a 400 MHz Bruker Avance III NMR spectrometer equipped with a 5-mm BBFO probehead, a Bruker Avance III 500 MHz NMR spectrometer equipped with a 5-mm nitrogen-cooled $^1$H, $^{13}$C, $^{15}$N Prodigy probehead and a Bruker Avance III HD 800 MHz NMR spectrometer, equipped with a cryogenically cooled, 5 mm $^1$H, $^{13}$C, $^{15}$N triple-resonance TCI probehead.

### HR-MS analyses

HR-MS analyses were performed on an Agilent 6560 ESI-IM-QTOF mass spectrometer equipped with AJS ESI ion source.

### UHPLC-MS analyses

UHPLC-MS analyses were performed on an Agilent 6530-QTOF mass spectrometer (ionisation mode: ESI +) equipped with an Agilent 1290 UHPLC inlet, UV detector and Autosampler. Eluents: solution A (99.9% water, 0.1 % trifluoroacetic acid), solution B (100% acetonitrile). Gradient: 0 min, 5% B; 1 min, 5% B; 10 min, 100% B. Flow rate: 0.5 mL/min.

### Melting Point

Melting points were measured on a Büchi Melting point B-540/B-545.

### Preparative HPLC

Water soluble compounds were purified by HPLC Shimadzu LC-8A system equipped with a Shimadzu array detector SPD-M20A using a Gemini 10 μm C18 column 110 Å, 100 × 21.20 mm 10 micron from Phenomenex. Eluents: solution A (99.9% water, 0.1% trifluoroacetic acid), solution B (100% acetonitrile). Gradient: 0 min, 5% B; 1 min, 5% B; 15 min, 100% B. Flow rate: 12 mL/min.

### DFT calculations

Cartesian coordinates for DFT minimized structures are provided in the file "Supplementary Data 1". Geometries of $(\mathbf{1})_2$, $(\mathbf{2})_2$, $(\mathbf{2})_3$, $(\mathbf{2})_3 \supset \mathbf{G2}$, and $(\mathbf{2})_3 \supset \mathbf{G4}$ were optimized without symmetry constraints. Initial structures were manually constructed and pre-optimized using GFN2-xTB. The final re-optimizations were performed at the BP86-D4-COSMO/def2-TZVP6-8 level of theory using the TURBOMOLE 7.7 suite with water as a solvent. Solvent effects were modeled using the COSMO model with default TURBOMOLE parameters. The D4 dispersion correction was included to account for crucial London dispersion

forces, which are essential for the stabilization of these supramolecular complexes, given the dominance of stacking interactions. Calculations of the helix cavity volume were performed using Swiss-PdbViewer.

## Data availability

The data supporting the findings of this study are available in the manuscript, in the Supplementary Information, or from the corresponding author. Cartesian coordinates for all DFT minimized structures are available in the file "Supplementary Data 1". The NMR data generated in this study have been deposited in the Figshare database under accession code 10.6084/m9.figshare.30579698.

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

## Acknowledgements

This work was supported by the Research Council of Finland (grant 357271 awarded to F.B.L.C.) and the University of Jyväskylä. We are grateful for project PID2023-148453NB-I00 funded by the Ministerio de Ciencia, Innovación y Universidades of Spain MCIU/AEI/10.13039/501100011033 and FEDER, UE (awarded to A.F.).

## Author contributions

D.D., R.A., and P.V. performed chemical synthesis. D.D. and P.P. performed NMR analysis. R.M.G. and A.F. performed computational work. F.B.L.C. conceptualized and supervised the project. The paper was written through contributions from all authors, and all authors have given approval to the final version of the paper.

## Competing interests

The authors declare no competing interests.
