## [Transparent Peer Review file · Nature Communications]

Programmable Assembly of Multistranded Helices in Water

Corresponding Author: Dr Fabien Cougnon

Version 0:

Reviewer comments:

Reviewer #1

(Remarks to the Author)
see attached pdf file

Reviewer #2

(Remarks to the Author)

Cougnon and coworkers report on a strategy to predict the assembly of water-soluble oligo arylethynyl molecules composed of phenyl and pyridinium rings. The accessible assemblies (double and/or triple helices) depends on the number of pyridinium rings on the ends, and the number of hydrophobic rings on the interior of the bolaamphiphilic molecules. The researchers present a cleaver and simple way to predict whether the molecules will form double and/or triple helices based on hydrophobic collapse and electrostatic repulsion between the charged pyridinium rings. Given the challenges of predicting/controlling higher-order assembly in foldamer molecules this study will be of high interest to the broad readership of Nature Communications. I found this manuscript to be interesting and well written. I recommend publication after minor revision. My two main concerns with the paper is generalizability of the approach for the assembly of other molecules besides areylethynyl systems may be over stated as well as what happens when the boundary limits of equation 3 are not whole integers.

See below for further comments:

Abstract – stating that the double helices are static structures implies that they are not in equilibrium even if that equilibrium is shifted towards the double helix. Is that really the case?

Page 3, sentence starting with Electrostatic repulsion may... - The stated electrostatic repulsion is not clear from figure 1b. Also "same face of the helix" isn't defined until a little later in the text.

Is there a preferred orientation of the terminal pyridinium ring? Does the methyl group prefer to point towards the interior or the exterior?

More details in the main manuscript would be welcome of how the energy diagrams were constructed.

Page 13 – the sentence in the second to last paragraph should read "...to bind more strongly to the triple helix.."

My main concerns are 1) that the predictable power of equation 3 may be a bit over stated. The manuscript doesn't show that this is a generalizable approach beyond arylethynyl foldamers.

2) The manuscript doesn't address what happens when the boundary limits (l/h and l/ρ) are not whole integers. For example, what happens when 4 hydrophobic or 4 polar aromatic rings are incorporated?

3) Does this approach hold true for other polar rings besides methyl pyridinium?

These concerns don't dampen my enthusiasm for the manuscript and I would like to see this work published in Nature Communications.

Reviewer #3

(Remarks to the Author)

In this manuscript, Cougnon and co-workers report on a new strategy to assemble multistranded foldamer assemblies. They make use of pyridinium repeat units at the ends of phenylene ethynylene oligomers. The number of pyridiniums restricts the potential modes of assembly as only certain combinations minimize electrostatic repulsion. Two oligomers are presented, giving different behavior, either a single double helix or a mixture of two interconverting double/triple helices. Each system has been thoroughly characterized by NMR spectroscopy in terms of concentration, temperature, and response to different guests.

Overall, I found this paper fascinating. The analysis at the beginning, developing predictions for assembly based on the relative lengths of charged/hydrophobic segments, is clever and a satisfying interpretation of the results. The experiments are thorough and support the conclusions. I believe that this work will be of interest to the foldamers community and more broadly to supramolecular chemists. I have only some very minor comments.

In Figure 2b and 3b, the authors present free energy profiles for assembly as a function of n , the number of strands in the assembly. I find these a little confusing. Since n can only assume integer values, it doesn't make sense to have a smooth curve in the plot. Also, the axis should probably be G° , not ΔG° , since it's not clear exactly what reactants/products are being compared, although H might make more sense since this is primarily based on electrostatic repulsion. Even so, I'm not sure we really know enough about the relative stabilities of, say, $n = 4$ and $n = 1$ to make a comparison. I suggest leaving these out.

On p 5, when discussing the DFT calculations, the manuscript says that "Single helix formation is disfavored, as it would necessarily bring charged pyridiniums in close contact, resulting in destabilizing electrostatic repulsion". It then goes on to explain that higher-order helices are also unfavorable. The statements are fine, but their placement in this paragraph implies (at least to me) that DFT calculations were performed on unassembled 1 and triple helix (1)3 to show that they are not possible. These calculations are not in the SI. If they have been done, this should be discussed; if not, this paragraph should be made more clear.

Is it correct to describe the behavior of oligomer 1 as "static" and 2 as "dynamic"? To me, these terms have implications in molecular motions and fluctuations that are presumably similar for all of the assemblies here. I would suggest "fully restricted/partially restricted", "unresponsive/responsive", "fixed/variable", or "fully constrained/partially constrained", as alternatives.

It seems like the data in Figure S24 could be used to extract thermodynamic parameters for the double helix \rightleftharpoons triple helix equilibrium, although perhaps the signals are too broad for reliable integration?

The SI includes very specific C-13 NMR assignments. A lot of these seem like they would be very difficult to do without HSQC or HMBC (or similar) experiments. In some cases the peaks are so close together I am surprised they could be done even with the help of these experiments. If these experiments were done, the data should be included. If not, the SI should explain how the assignments were made (or just leave them out, as they are not necessary).

There is a typo in Fig. 4: OTF instead of OTf.

Version 1:

Reviewer comments:

Reviewer #1

(Remarks to the Author)

After reviewing the manuscript once more and carefully reading the authors' detailed and honest responses to the questions raised, I have no further concerns that would justify delaying the publication of this nice, well-documented and clearly presented piece of work.

Reviewer #2

(Remarks to the Author)

The authors have adequately addressed my comments and I recommend publication.

Reviewer #1 (Remarks to the Author):

In this work, the group of Fabien Cougnon reports the reengineering by computational design of a supramolecular scaffold developed in the late 1990 (Moore et al.) which has recently regained interest (Berryman et al.). The main objective here was to study the formation of multistranded helices and their application as supramolecular hosts for mono- or di-ionic molecules.

The oligo(m-phenylene ethylene) scaffold was therefore selected and chemically modified at both extremities to yield bola-amphiphilic strands. These oligomers were designed and modelled to fold into double or triple helices, depending on an inequality relating the length and positioning of positively charged segments to the hydrophobic core.

Two m-phenylene ethylene-containing sequences were synthesized in solution in satisfying yields and were well characterized by NMR and ESI-MS spectroscopies. Both synthesized strands were composed of N-methylpyridinium residues (3 and 2 for compound 1 and 2 respectively) on both ends and a constant m-phenylene ethylene segment (3 units).

For both strands, the ¹H NMR spectra showed only one single species in CD₃CN (moderately polar solvent), while moving to pure water new species emerged with variable proportions according to different parameters like concentration, temperature and counterions (TFA versus OTf).

The structural characterization of the new helical folds was thoroughly performed by 2D NMR ROESY and DOSY experiments, which confirmed in the case of the strand 1 the formation of a stable double helix in water, when isolated as a TFA salt. For compound 2, the authors observed that the OTf salt was showing an unfolded strand in CD₃CN and double or triple helix in a concentration- and temperature-dependent manner. Each structural result and conclusion was accompanied by a detailed description and interpretation of the NMR data recorded in solution.

Subsequently, the authors investigated the potential of these new supramolecular architectures as host for ionic guests. In particular, they demonstrated that strand 2 underwent a dynamic switch from the double to the triple helix upon binding to guest G2. Once bound in the cavity of the triple helix, G2 stabilized it and delay its transition to a double helix upon sample dilution or heating.

The manuscript is overall very well-written, and as stated above, the results are very well-described with proof of evidences, detailed protocols and thorough NMR structural characterization.

Nonetheless, several points and comments should be addressed prior to acceptance for publication.

To improve the readability and facilitate quick access to the supporting figures, all figures should be presented at the beginning of the supporting document, and appeared in the order in which they are cited throughout the manuscript.

We thank the reviewer for this helpful comment. Upon reviewing the manuscript and Supplementary Information, we realized that the order of several supporting figures was indeed not optimal. The Supplementary Information has been reorganized to better follow the narrative of the manuscript, and all references to supplementary figures have been updated to ensure that each figure is properly cited in the manuscript.

A separate section should describe the synthesis part of compounds 1 and 2.

As suggested by the reviewer, the syntheses of strands 1 and 2 have been grouped into a dedicated section at the beginning of the Supplementary Information (“2. Synthesis of the strands”).

Moreover, all supporting figures should be referenced if not discussed in the manuscript, even if only briefly. For instance, Figure S6 - which appears to represent a key experiment- is not mentioned at all, which seems to be a clear oversight. This NMR experiment enables monitoring the transition from an unfolded strand to a double helix by varying the proportion of water in the deuterated solvent.

Every important supplementary figure is now properly cited in the manuscript. In the case of Fig. S6 (now Fig. S7), for example, the text now reads:

“Substantial spectral changes, indicative of duplex formation, were observed upon addition of D₂O (Supplementary Fig. S7).”

In the Figure 2, showing the two DOSY spectra of the folded and unfolded strand 1. TFA might help in visualizing the differences in the hydrodynamic radius.

DOSY spectra recorded under identical conditions (same solvent and temperature) are indeed often compared directly to visually highlight differences in the size of supramolecular assemblies. In our case, however, the DOSY spectra of the unfolded strand and the corresponding double helix were recorded in different solvents (acetonitrile and water, respectively). The diffusion coefficients visible in the spectra cannot therefore be directly compared. Instead, hydrodynamic radii must be calculated to allow a meaningful comparison. Presenting the spectra side by side could thus be misleading. For this reason, we prefer to keep the figure in its current form.

The second main concern is about the concentrations used for the NMR experiments. Firstly, they should be mentioned in all the captions of the figures reporting NMR data.

The concentrations have been added to all figure captions where they were previously missing. For example, in the caption of Fig. 2:

“**d**, ¹H NMR spectrum of unfolded strand **1**·**TFA** in CD₃CN (5 mM, 500 MHz, 298 K).”

And Fig. 4:

“**d**, ¹H NMR spectrum of unfolded strand **2**·**OTf** in CD₃CN (10 mM, 500 MHz, 298 K).”

Secondly, how the authors explain the variations in the concentration of their NMR samples. For instance, in the case of the double helix of **1**·**TFA** in deuterated water the concentration was set at 10 mM - twice the concentration for the monomeric strand in CD₃CN - which appears logical. However, upon switching to OTf salts, the concentration was reduced to 3 mM, and two set of signals were observed (Figure S5). The second set of peaks was attributed to the binding of the OTf ions within the cavity of the double helix. However, in the absence of the **1**·**OTf** single strand spectrum in CD₃CN, one could also hypothesized that the reported spectrum in Figure S5 reflects a mixture between the unfold state and the double helix of **1**·**OTf**.

There was a mistake in the caption of this figure (now Supplementary Fig. S13): the NMR spectra of the unfolded strand, **(1)₂·TFA** and **(1)₂·OTf** were all recorded at 5 mM. The caption was therefore corrected with these concentrations.

In panels b) and c), only a single set of signals is observed. These signals, highlighted in orange, were assigned to the double helices **(1)₂·TFA** and **(1)₂·OTf**. The spectrum of **(1)₂·OTf** differs only slightly from that of **(1)₂·TFA**, with small chemical shift changes attributed to triflate binding (as further confirmed by the titration shown later in Supplementary Fig. S15).

The figure has been modified for improved clarity:

In the text, they authors mentioned that both couterions showed the same behaviour by NMR spectroscopy but the 1H NMR spectrum of $1\cdot\text{OTf}$ is not included in the Figure S5.

The original sentence has been revised to match more accurately the content of Supplementary Fig.13 (previously Supplementary Fig. S5), which compares the ^1H NMR spectra of $(1)_2\cdot\text{TFA}$ and $(1)_2\cdot\text{OTf}$:

“Duplex formation was also found to be independent of the counterions tested (trifluoroacetate or triflate). However, subtle differences in the NMR spectra of $(1)_2\cdot\text{TFA}$ and $(1)_2\cdot\text{OTf}$ (Supplementary Fig. S13) revealed that the double helix interacts differently with these two counterions.”

There appears to be an error in the caption of the Figure 3. It refers to strand 2, whereas the subsequent paragraph clearly discussed strand 1.

This mistake has been corrected. The caption now reads: “**c-e**, Partial ^1H NMR spectra of double helix $(1)_2\cdot\text{TFA}$ in D_2O (5 mM in D_2O , 500 MHz, 298 K)”.

In the sections dealing with the use of the double or triple helix cavity to accommodate small molecules, the authors have to define the volume of these cavities, to give an idea of their relative size.

This can be presumably extracted from the modelling performed to validate the design and strand construction.

We agree that these values are important for providing a better appreciation of the cavity size. The cavity volumes were calculated using Swiss-PdbViewer, as described on page S63 of the Supplementary Information, and are summarized in Table S1:

Calculations of the helix cavity volume were performed using Swiss-PdbViewer.^{S11} Since the helices do not form fully enclosed cavities, we followed the standard procedure for open cavities by capping the ends with a triethynylbenzene unit to define the volume boundaries.

Table S1 | Cavity volumes of multistranded helices calculated with Swiss-PdbViewer.

Helix	Volume (Å ³)	Interior volume (Å ³)
Double helix (1) ₂	3022	378
Double helix (2) ₂	2314	222
Triple helix (2) ₃	3470	469

Figure S51 | Cavity volume of triple (2)₃ capped with a triethynylbenzene molecule.

The volumes are also mentioned in two places in the manuscript itself:

- “We hypothesized that longer, more hydrophobic anions (**G2-G4**) could bind more strongly by threading through the helix cavity (internal volume: 378 Å³).”
- “This increased affinity is attributed to the longer and more hydrophobic internal cavity of the triple helix (2)₃ (internal volume: 469 Å³), which provides a more favorable environment for guest encapsulation.”

In the figures S1 and S15, the authors report the UPLC traces of final compounds 1 and 2 eluting at 4 and 4.5 min respectively. However, there is no mention of the UPLC conditions (column, solvent, gradient) used.

This information has been added to the section “Methods and General Information” of the Supplementary Information:

“**UHPLC-MS analyses.** UHPLC-MS analyses were performed on an Agilent 6530-QTOF mass spectrometer (ionisation mode: ESI+) equipped with an Agilent 1290 UHPLC inlet, UV detector and

Autosampler. Eluents: solution A (99.9% water, 0.1% trifluoroacetic acid), solution B (100% acetonitrile). Gradient: 0 min, 5% B; 1 min, 5% B; 10 min, 100% B. Flow rate: 0.5 mL/min.”

According to the conditions (higher % of water with respect to acetonitrile), is there any possibility that the peaks on the chromatograms do correspond to the double helix of 1 and 2?

The reviewer raises an excellent point. However, the multistranded helices are unlikely to remain intact under UHPLC conditions. First, the sample ($C = 0.1 \text{ mM}$, $V_{\text{injection}} = 4 \text{ }\mu\text{L}$) is immediately diluted in the eluent upon injection. Second, the compounds elute when the solvent gradient reaches 50–60% acetonitrile. Solvent-dependent ^1H NMR experiments (Figs. S7 and S21) indicate that the helices are essentially dissociated when the acetonitrile content raises above 50%. Both the low effective concentration and the high proportion of acetonitrile promote helix disassembly. The MS spectra obtained in tandem with the UHPLC chromatography (Supplementary Figs. S1 and S2) show only the presence of monomeric strands, which could support the conclusion that the helices are dissociated under these conditions.

In this same line, did they authors attempt to characterize their double and triple helix by ESI-MS analyses.

We attempted to characterize both double helix (1)₂ and triple helix (2)₃ by ESI-MS and ion mobility mass spectrometry. This experiment proved challenging because it required the addition of an organic co-solvent (acetonitrile or methanol) to the aqueous solution of the multistranded helices to enable proper vaporization upon direct injection into the MS source.

Initial results obtained with strand 2 were promising (Figure 1), as we were able to identify an ion peak corresponding to the expected trimeric species. However, the intensity of this peak was low, and these data must be interpreted with caution, since organic compounds frequently fly as dimeric or trimeric aggregates. Therefore, the detection of a trimeric species does not necessarily indicate the triple helix survives in the gas phase.

Chemical Formula: $\text{C}_{54}\text{H}_{38}\text{N}_4^{4+}$
Molecular Weight: 742,9238

Ion	z	m/z_{exp}	m/z_{theor}	MW (Da)	mass accuracy (mDa)	DT (ms)	Height	$^{DT}\text{CCS}_{\text{N}_2}$ (\AA^2)
	2	520,108		1040,216		26,51		
	1	703,1514		703,1514				
	1	1025,1902		1025,19				
$[\text{M}+3\text{OTf}]^+$	1	1189,1686	1189,1652	1189,169	-3,4	47,69	19880752	351
$[\text{2M}+6\text{OTf}]^{2+}$	2	1189,1649	1189,1652	2378,33	0,3	33,84	2451558	491,6
	1	1527,0328		1527,033				367,2
	2	1528,031		3056,062				555,3
$[\text{3M}+10\text{OTf}]^{2+}$	2	1859,2278	1859,2262	3718,456	-1,6	41,35	2458753	600,6

Figure 1. Observed ions, mass accuracies, drift times and $^{DT}\text{CCS}_{\text{N}_2}$ values with strand 2.

Attempts to optimize the experimental conditions to improve data quality were unsuccessful. Moreover, the same experiment performed on an aqueous solution of the double helix (**1**)₂ yielded multiple peaks corresponding to both dimeric and trimeric species (Figure 2). Given that strand **1** is unlikely to form a triple helix, even in the gas phase, these results suggest that the observed oligomeric species were artefacts arising from aggregation during sample vaporization. Consequently, these data were not included in the manuscript, and further MS characterization of the helices was not pursued.

Ion	z	m/z_{exp}	m/z_{theor}	MW (Da)	mass accuracy (mDa)	DT (ms)	Height	$^{\text{DT}}\text{CCS}_{\text{N}_2}$ (\AA^2)
	2	457,1599		914,3198				
	2	539,1474		1078,295				
	1	701,979		701,979				
	2	703,1202		1406,24				
	2	703,1215		1406,243				
$[\text{M}+4\text{OTf}]^{2+}$	2	785,1087	785,1083	1570,217	-0,4	28,66	946540	416,8
	2	1556		3112				
	2	1638,1842		3276,368				
$[\text{M}+5\text{OTf}]^+$	1	1719,171	1719,1693	1719,171	-1,7	56,08	186115	409
$[2\text{M}+10\text{OTf}]^{2+}$	2	1720,1729	1720,1712	3440,346	-1,7	40,59	513876	589,8
$[3\text{M}+15\text{OTf}]^{3+}$	3	1720,1755	1720,1711	5160,527	-4,4	30,58	104092	757,6

Figure 2. Observed ions, mass accuracies, drift times and $^{\text{DT}}\text{CCS}_{\text{N}_2}$ values with strand **1**.

Figure S38 scheme is misleading and should be improved.

There was an error in this figure, where the triple helix was shown twice. The figure (now Fig. S40) has been corrected to show the triple helix bound to guest **G4**, which is consistent with the NMR spectra shown below.

Finally, if the purpose of this work was to study the dynamic of their supramolecular objects in aqueous medium, it is unclear why the authors did not investigate the behaviour of these multistranded helices as HCl salts? This choice could have facilitated crystallization – an aspect not addressed in this manuscript, and potentially yielded valuable structural insights.

The trifluoroacetate and triflate salts were chosen because they allowed us to investigate thoroughly the assembly of different multistranded helices in solution. The chloride salts of both strands **1** and **2** were also synthesized with the specific aim of obtaining crystals suitable for X-ray diffraction.

However, due to the challenging nature of the synthesis of these compounds, only small amounts (approximately 2 mg of each strand) were obtained. These samples were sent to the group of Prof. Ivan Huc, who has extensive experience in obtaining diffracting crystals of foldamers from limited material. This study is still ongoing, and despite several months of effort, no suitable crystal structure has yet been obtained.

Given the inherent unpredictability of this crystallographic work, and the fact that we have already initiated follow-up studies on the functionalization of these helices, we decided to submit this study supported by DFT calculations. Accordingly, the synthesis of the chloride salts is not described in the current manuscript. If crystal structures are eventually obtained, they will be published in collaboration with Prof. Huc in due course.

In conclusion, once the authors have clarified the points listed above and improved the supporting information, the manuscript will merit publication in Nature Communications.

Reviewer #2 (Remarks to the Author):

Cougnon and coworkers report on a strategy to predict the assembly of water-soluble oligo arylethynyl molecules composed of phenyl and pyridinium rings. The accessible assemblies (double and/or triple helices) depends on the number of pyridinium rings on the ends, and the number of hydrophobic rings on the interior of the bolaamphiphilic molecules. The researchers present a clever and simple way to predict whether the molecules will form double and/or triple helices based on hydrophobic collapse and electrostatic repulsion between the charged pyridinium rings. Given the challenges of predicting/controlling higher-order assembly in foldamer molecules this study will be of high interest to the broad readership of Nature Communications. I found this manuscript to be interesting and well written. I recommend publication after minor revision. My two main concerns with the paper is generalizability of the approach for the assembly of other molecules besides areylethynyl systems may be over stated as well as what happens when the boundary limits of equation 3 are not whole integers.

These concerns are restated and addressed in detail below.

See below for further comments:

Abstract – stating that the double helices are static structures implies that they are not in equilibrium even if that equilibrium is shifted towards the double helix. Is that really the case?

We agree with the reviewer's comment. We have removed the word "static" from the abstract and replaced it with a more accurate description:

"We demonstrate that the sequence of oligo(*m*-phenylene ethynylene) strands composed of hydrophobic phenylene and charged pyridinium residues reliably direct the formation of either a single structure (e.g., a double helix) or dynamic assemblies (e.g., double and triple helices in exchange)."

Page 3, sentence starting with Electrostatic repulsion may... - The stated electrostatic repulsion is not clear from figure 1b. Also "same face of the helix" isn't defined until a little later in the text.

Figure 1 and the corresponding text have been modified to improve clarity:

“Here we present an approach for encoding, within the primary sequence of synthetic strands, all the information necessary to control both the assembly and dynamic behavior of multistranded helices. Fig. 1a shows the general structure of a bolaamphiphilic strand, consisting of a central hydrophobic core flanked by two permanently charged, polar termini. Let h and p denote the lengths of the hydrophobic and polar segments, respectively. Such bolaamphiphilic sequences promote the self-assembly of n -stranded helices ($n = 1, 2, 3, \dots$) in aqueous environments. As illustrated in Fig. 1b, the resulting helices feature a central hydrophobic layer flanked by two charged layers at the upper and lower rims, effectively minimizing exposure of the hydrophobic surface to water.

The formation of these helices occurs only if the charged segments do not stack directly above one another, as this would generate destabilizing electrostatic repulsion. Repulsion between charged segments located on the upper rim can be avoided if the combined length of these charged segments ($n \cdot p$) does not exceed the length of one full helical turn (Fig. 1b, inset). Defining l as the length of strand required to complete one helical turn, this condition can be expressed as:

$$n \cdot p \leq l \quad (1)$$

By symmetry, this same constraint applies to the charged segments located at the lower rim of the helix.

Repulsion between charged segments located on opposite rims of the helix can be similarly avoided if they are separated by a full hydrophobic layer; that is, if the combined length of hydrophobic components located at the center of the structure ($n \cdot h$) is greater or equal to l :

$$n \cdot h \geq l \quad (2)$$

Combining inequalities (1) and (2) yields a single inequality that defines the permissible range for the number of strands n :

$$l/h \leq n \leq l/p \quad (3)$$

Is there a preferred orientation of the terminal pyridinium ring? Does the methyl group prefer to point towards the interior or the exterior?

The terminal pyridinium ring does not adopt a preferred orientation: the methyl group can point either outward or inward, as evidenced by ROESY spectroscopy (Supplementary Figs. S9 and S27). The rotation of the terminal pyridinium residue is explicitly showed in panel b) of Supplementary Figs. S9 and S27, and this behavior is noted in the corresponding figure captions:

More details in the main manuscript would be welcome of how the energy diagrams were constructed.

We agree with the reviewer that the original description of these diagrams was too brief and could lead to confusion. To clarify their purpose and construction, a paragraph has been added to the manuscript explaining the qualitative nature of these energy diagrams:

“This situation can be schematically represented with a diagram (Fig. 2b) comparing the relative enthalpies of three possible states: $n = 1$ (single helix), $n = 2$ (double helix), and $n = 3$ (triple helix). The double helix is more stable (lower enthalpy) than the single and triple helices (higher enthalpy) because

it is the only configuration that avoids electrostatic repulsion. It should be noted that these enthalpic levels are arbitrary and intended solely as a qualitative illustration of the allowed and forbidden states.”

Page 13 – the sentence in the second to last paragraph should read “...to bind more strongly to the triple helix..”

The sentence has been corrected accordingly.

May main concerns are 1) that the predictable power of equation 3 may be a bit over stated. The manuscript doesn't show that this is a generalizable approach beyond arylethynyl foldamers.

Two discussions of the model's limitations have been added to the manuscript: the first immediately following the introduction of the model, and the second in the conclusion.

- 1) “In this article, we sought to experimentally test the predictions of the model described above. However, real molecular systems do not necessarily meet the assumptions of this idealized framework, which presumes a backbone of constant thickness, sufficient flexibility to accommodate helices of any multiplicity, and uniformly distributed charges across the polar termini. Testing the validity of our model therefore required choosing a molecular scaffold that approximates these idealized features as closely as possible.”
- 2) In the conclusion: “This study establishes a robust strategy for controlling both the assembly and dynamic behavior of multistranded helices through sequence design in oligo(*m*-phenylene ethynylene) strands. The key strength of this approach lies in its minimalism: precise structural control was achieved using only two building blocks, hydrophobic phenylene and charged pyridinium residues. This simplicity renders the approach modular and potentially extensible to alternative sequence patterns, including non-bolaamphiphilic ones. The strategy may also be applied to strands containing anionic residues (instead of cationic ones) and to other polyaromatic backbones (such as *ortho*-phenylene ethynylene and related *ortho*- or *meta*-phenylene frameworks) provided they possess sufficient flexibility to form multistranded helices. Moreover, the oligo(*m*-phenylene ethynylene) backbone is amenable to chemical modification,^{38,41} allowing for the integration of catalytic, recognition, or sensing functions within structurally programmed helices.”.

2) The manuscript doesn't address what happens when the boundary limits (l/h and l/p) are not whole integers. For example, what happens when 4 hydrophobic or 4 polar aromatic rings are incorporated?

This situation was not discussed because inequality (3) applies regardless of whether the boundary values are integers. In the case of an oligophenylene composed of four hydrophobic residues and four pyridinium residues ($l = 6$, $h = 4$, and $p = 4$), the two boundary values are $l/h = l/p = 1.5$. Since the number of strands in a helix can only take integer values ($n = 1, 2, 3, \dots$), the inequality $1.5 \leq n \leq 1.5$ cannot be satisfied, and therefore no helix can form. This can be readily visualized using a schematic representation of the relative positions of the aromatic units (phenylenes shown in yellow and pyridiniums in blue) in single-, double-, and triple-helical arrangements, which illustrates that electrostatic repulsion cannot be avoided in this case:

3) Does this approach hold true for other polar rings besides methyl pyridinium?

Because our approach relies on electrostatic repulsion to control the formation of specific helical architectures, the polar residues need to remain permanently charged. In principle, these residues may carry either positive or negative charges. The paragraph we have added to the conclusion now mentions this alternative possibility:

“This study establishes a robust strategy for controlling both the assembly and dynamic behavior of multistranded helices through sequence design in oligo(*m*-phenylene ethynylene) strands. The key strength of this approach lies in its minimalism: precise structural control was achieved using only two building blocks, hydrophobic phenylene and charged pyridinium residues. This simplicity renders the approach modular and potentially extensible to alternative sequence patterns, including non-bolaamphiphilic ones. The strategy may also be applied to strands containing anionic residues (instead of cationic ones) and to other polyaromatic backbones (such as *ortho*-phenylene ethynylene and related *ortho*- or *meta*-phenylene frameworks) provided they possess sufficient flexibility to form multistranded helices. Moreover, the oligo(*m*-phenylene ethynylene) backbone is amenable to chemical modification,^{38,41} allowing for the integration of catalytic, recognition, or sensing functions within structurally programmed helices.”

These concerns don't dampen my enthusiasm for the manuscript and I would like to see this work published in Nature Communications.

Reviewer #3 (Remarks to the Author):

In this manuscript, Cougnon and co-workers report on a new strategy to assemble multistranded foldamer assemblies. They make use of pyridinium repeat units at the ends of phenylene ethynylene oligomers. The number of pyridiniums restricts the potential modes of assembly as only certain combinations minimize electrostatic repulsion. Two oligomers are presented, giving different behavior, either a single double helix or a mixture of two interconverting double/triple helices. Each system has been thoroughly characterized by NMR spectroscopy in terms of concentration, temperature, and response to different guests.

Overall, I found this paper fascinating. The analysis at the beginning, developing predictions for assembly based on the relative lengths of charged/hydrophobic segments, is clever and a satisfying

interpretation of the results. The experiments are thorough and support the conclusions. I believe that this work will be of interest to the foldamers community and more broadly to supramolecular chemists. I have only some very minor comments.

In Figure 2b and 3b, the authors present free energy profiles for assembly as a function of n , the number of strands in the assembly. I find these a little confusing. Since n can only assume integer values, it doesn't make sense to have a smooth curve in the plot. Also, the axis should probably be G° , not ΔG° , since it's not clear exactly what reactants/products are being compared, although H might make more sense since this is primarily based on electrostatic repulsion. Even so, I'm not sure we really know enough about the relative stabilities of, say, $n = 4$ and $n = 1$ to make a comparison. I suggest leaving these out.

We fully agree with the reviewer, and Figures 2b and 3b have been modified accordingly. The energy diagrams now display discrete levels for each state, and the axis label (ΔG) has been replaced with enthalpy.

As mentioned earlier, these diagrams are intended solely as visual illustrations of the allowed and forbidden states. For this reason, we have retained the forbidden states ($n = 1$ and 4), even though the exact enthalpy values were not calculated for each state. We hope that the explanations added to the manuscript, as requested by Reviewer 2, have clarified both the purpose and the limitations of this representation.

On p 5, when discussing the DFT calculations, the manuscript says that "Single helix formation is disfavored, as it would necessarily bring charged pyridiniums in close contact, resulting in destabilizing electrostatic repulsion". It then goes on to explain that higher-order helices are also unfavorable. The statements are fine, but their placement in this paragraph implies (at least to me) that DFT calculations were performed on unassembled 1 and triple helix (1)₃ to show that they are not possible. These calculations are not in the SI. If they have been done, this should be discussed; if not, this paragraph should be made more clear.

We agree with the reviewer that the mention of DFT calculations in this paragraph was ambiguous, as modeling was performed only for the double helix. To resolve this ambiguity, we have modified this section as follows:

"The equilibria involved in the folding and assembly of strand **1** are shown in Fig. 2c. According to our predictions, single helix formation is disfavored, as it would bring charged pyridiniums in close contact, resulting in destabilizing electrostatic repulsion. The formation of a double helix (which was modeled using a BP86-D4-COSMO(water)/def2-TZVP level of theory) allows the outer pyridiniums to effectively shield the hydrophobic phenylenes from water. The assembly of higher-order helices (not shown in Fig. 2c) is also prohibited, since the pyridinium residues already occupy all available positions at both the top and the bottom of the double helix, making it impossible for additional charged residues to fit without overlapping."

Is it correct to describe the behavior of oligomer 1 as "static" and 2 as "dynamic"? To me, these terms have implications in molecular motions and fluctuations that are presumably similar for all of the assemblies here. I would suggest "fully restricted/partially restricted", "unresponsive/responsive", "fixed/variable", or "fully constrained/partially constrained", as alternatives.

The terms "static" and "dynamic" may not be the most appropriate in this context. Since the distinction between the two systems is already described in detail in the manuscript, we have removed references to "static behavior" and "dynamic behavior".

It seems like the data in Figure S24 could be used to extract thermodynamic parameters for the double helix \rightleftharpoons triple helix equilibrium, although perhaps the signals are too broad for reliable integration?

We also considered that quantitatively evaluating the relative ratios of the double and triple helices in concentration- and temperature-dependent experiments could provide valuable insights into the thermodynamics of the exchange process. Unfortunately, the NMR signals could not be consistently integrated, either because they were too broad or overlapped with other resonances.

In the experiment described in Supplementary Fig. S24 (now Supplementary Fig. S30), the double-to-triple helix transition was performed in the presence of triflate, which can interact with both the double and triple helices with different affinities. The participation of the counterion in the equilibria associated with the transition must therefore be considered for any quantitative interpretation of the data. Consequently, even if the NMR signals could be accurately integrated, obtaining meaningful thermodynamic parameters would not be straightforward and would require a more sophisticated model than the simple $(2)_2 \rightleftharpoons (2)_3$ equilibrium.

The SI includes very specific C-13 NMR assignments. A lot of these seem like they would be very difficult to do without HSQC or HMBC (or similar) experiments. In some cases the peaks are so close together I am surprised they could be done even with the help of these experiments. If these experiments were done, the data should be included. If not, the SI should explain how the assignments were made (or just leave them out, as they are not necessary).

As correctly noted by the reviewer, the assignment of the ^{13}C NMR spectra was performed using HSQC and HMBC experiments. These experiments have now been added to the Supplementary Information for both strands **1** (Supplementary Fig. S5) and **2** (Supplementary Fig. S20). Most of the 2D correlations are sufficiently well resolved to enable precise assignment of the ^{13}C signals. In a few cases where signals are too close to be distinguished, they have been labeled with the corresponding list of carbons (e.g., two overlapping peaks at ca. 91 ppm are labeled “29,6” in Fig. S4).

There is a typo in Fig. 4: OTF instead of OTf.

This typo has been corrected.